# Morin Treatment Delays the Ripening and Senescence of Postharvest Mango Fruits

**DOI:** 10.3390/foods12234251

**Published:** 2023-11-24

**Authors:** Lihong Guo, Kaiqi Liang, Xiaochun Huang, Weiqian Mai, Xuewu Duan, Fuwang Wu

**Affiliations:** 1School of Food Science and Engineering, Foshan University, Foshan 528200, China; guolihong8184@163.com (L.G.); 13542632699@163.com (K.L.); 17728432549@163.com (X.H.); 2Guangdong Provincial Key Laboratory of Food Intelligent Manufacturing, Foshan 528200, China; maiweiqian@foxmail.com; 3Guangdong Provincial Key Laboratory of Applied Botany, South China Botanical Garden, Chinese Academy of Sciences, Guangzhou 510650, China; xwduan@scbg.ac.cn

**Keywords:** morin, mango, ripening and senescence, quality deterioration, gene expression

## Abstract

A 0.005% and 0.01% morin treatment was applied to treat mango fruits stored under ambient conditions (25 ± 1 °C) with 85–90% relative humidity, and the effects on quality indexes, enzyme activity related to antioxidation and cell wall degradation, and gene expressions involved in ripening and senescence were explored. The results indicate that a 0.01% morin application effectively delayed fruit softening and yellowing and sustained the nutritional quality. After 12 days of storage, the contents of soluble sugar and carotenoid in the treatment groups were 68.54 mg/g and 11.20 mg/100 g, respectively, lower than those in control, while the vitamin C content in the treatment groups was 0.58 mg/g, higher than that in control. Moreover, a morin application successively enhanced the activity of superoxide dismutase (SOD), catalase (CAT), and peroxidase (POD), but reduced the activity of polygalacturonase (PG) and pectin lyase (PL). Finally, real-time PCR and correlation analysis suggested that morin downregulated the ethylene biosynthesis (*ACS* and, *ACO*) and signal transduction (*ETR1*, *ERS1*, *EIN2*, and *ERF1*) genes, which is positively associated with softening enzymes (*LOX*, *EXP*, *βGal*, and *EG*), carotenoid synthesis enzymes (*PSY* and, *LCYB*), sucrose phosphate synthase (*SPS*), and uncoupling protein (*UCP*) gene expressions. Therefore, a 0.01% morin treatment might efficiently retard mango fruit ripening and senescence to sustain external and nutritional quality through ethylene-related pathways, which indicates its preservation application.

## 1. Introduction

Mango (*Mangifera indica* L.) fruits originate in tropical regions and are highly praised worldwide for their rich nutrition [1]. Mango fruits have been reported to help reduce liver cholesterol deposition, prevent constipation, promote intestinal health, and demonstrate anti-inflammatory effects [2]. However, as a climacteric fruit, postharvest mango fruits still carry on a vigorous respiratory metabolism, accompanied by a decline in hardness, color change, an increase in soluble sugar content, and carotenoid synthesis, and they gradually undergo ripening and quality deterioration after being stored at room temperature for 7–9 days [3]. In addition, because of the high temperatures and rainy conditions during its development stage, mango is vulnerable to microbial damage, which causes a lot of decay in postharvest storage and transportation. The primary disease that causes mango postharvest rot in all production areas worldwide is anthracnose, caused by the pathogenic fungus *Coccidiomycete* in the asexual phase [4]. With the continuous expansion of mango cultivation and the increase in yield year by year, the mango industry has become an economic pillar in many countries. However, anthracnose can cause a 30−60% yield loss of mango every year, which has restricted the development of the mango industry [5].

To date, many physical and chemical methods have been widely applied to delay mango ripening and prevent postharvest diseases in various countries. For example, heat treatment is one of the most commonly used postharvest fruit preservation technology methods. Studies have shown that treatment at 60 °C for 1 min can not only reduce the relative conductivity, respiration rate, and malonaldehyde content of postharvest mangoes but can also elevate the total phenol and flavonoid content, thereby resulting in quality maintenance during storage [6]. In coating preservation, a chitosan/titanium dioxide nanocomposite (nano-TiO_2_) coating can effectively inhibit the growth and infestation of pathogens during the storage of mangoes, decrease the fruit decay rate, and maintain fruit quality [7]. Additionally, adding Thai essential oils to hydroxypropyl methylcellulose (HPMC) in nanocomposite coatings can have a preservation effect on mangoes [8]. In addition, modified atmosphere packaging (MAP) and essential oil (EO) vapor treatment can also effectively control the postharvest decay of mangoes, extending their storage period and shelf life [9]. Although these methods can retard the maturation and senescence of postharvest mangoes, their application still has limitations, such as being of high cost, time consuming, and chemically hazardous. In recent years, an increasing number of people have focused on food safety issues, such as production costs, residual toxicity, and environmental pollution during food processing and preservation. Therefore, developing a green, efficient, and low energy consumption mango preservation technology is of practical significance.

Flavonoids can enhance the activity of antioxidation and free radical scavenging in vivo. Studies have shown that the hydrogen atoms on phenolic hydroxyl groups can combine with superoxide radicals to generate flavonoid free radicals, which can stop the free radical chain reaction by reacting with other free radicals [10]. Moreover, abiotic stresses, such as drought, can promote flavonoid synthesis and accumulation in plants [11], which can prevent plant damage caused by viruses, fungi, bacteria, and herbivores [12]. Furthermore, a recent study revealed that ‘Laoyaban’ flavonoids also have a particular preservative effect on ‘Hongyan’ strawberries through their antibacterial and antioxidant activities [13]. Therefore, flavonoids, as natural plant metabolites, could potentially be developed as a broad-spectrum and safe preservative. Morin is a yellow polyhydroxy flavonoid compound extracted from mulberry plants, and it possesses antioxidant, anti-inflammatory, anti-tumor, anti-diabetic, anti-hypertension, antibacterial, uric acid reduction, and nerve protection functions [14]. However, current studies mainly concentrate on the functional activity of morin in mammals, with few reports on postharvest fruits such as the mango. Furthermore, a former study illustrated that morin treatment could delay the degradation of water-soluble polysaccharides in postharvest bananas [15], but the mechanism by which morin delays fruit ripening and senescence is yet to be explored, especially in molecular biology. Therefore, it is worth revealing the biological function of morin in delaying the postharvest quality deterioration of mango fruits.

## 2. Materials and Methods

### 2.1. Fruit Materials and Treatments

Mango fruits (*Mangifera indica* L. cv. ‘Zill’) at the mature green stage were harvested from an orchard (Siyuan Farm) in Shishan Town, Nanhai District, Foshan City, Guangdong Province (E 112°59′59.54″, N 23°08′34.62″). The harvested mango fruits were immediately transported to the laboratory for processing within 1 h via a cargo van with air conditioning at 25 °C. Over 200 fruits that were free from insect pests, mechanical damage, and of uniform size were selected and randomly divided into two groups. Each group consisted of three replicates, with each replicate containing 27 fruits. The control fruits were then soaked in water for 3 min, whereas the experimental group fruits were soaked in a 0.01% morin solution (morin was purchased from Macklin, CAS: 654055-01-3). In a preliminary experiment of this study, 0.01% morin treatment presented a better preservation effect than 0.005% morin treatment (Appendix A). After natural drying at (25 ± 1 °C), groups of nine fruits were together placed in a 0.03 mm polyethylene film bag and stored at ambient temperature (25 ± 1 °C) with 85–90% relative humidity. Fruit hardness and flesh chroma were measured and photographed every 3 d during storage. The fruit pulp along the equator line was frozen with liquid nitrogen and placed in −80 °C refrigerators for further physiology and gene expression analysis.

### 2.2. Determination Method

#### 2.2.1. Measurement of Chromatic Aberration

The chromatic aberration measurement was conducted referring to Diop et al.’s method [16], with appropriate modifications. The flesh color difference indexes of the fruit along the equator line, namely, luminance (L*), chroma (C*), and hue angle (H*), were detected with a colorimeter (CR-10, Konica Minolta, Tokyo, Japan). Nine fruits were measured in each group, and the average value was calculated.

#### 2.2.2. Assay of Firmness

The firmness was tested according to the method of Teixeira et al. [17] with mild modifications. Nine fruits from each group were taken for a firmness analysis with a fruit firmness meter (GS-701G, Teclock, Nagano Prefecture, Japan) at three positions around the fruit equator line with equal spacing. The average value was calculated, and the results are expressed as Newton (N).

#### 2.2.3. Analysis of Soluble Sugar Content

The soluble sugar content was assayed using the method of Hor et al. [18], with mild modifications. After liquid nitrogen grinding, the mango powder sample (1.0 g) and 15 mL of deionized water were blended in an 80 °C water bath for 20 min and later centrifuged (4000× *g*, 10 min, 25 °C). The supernatant was adjusted to 50 mL, followed by dilution five times. Then, 0.1 mL diluent sample, 0.9 mL deionized water, and 5 mL anthrone reagent were mixed and boiled for 10 min. The absorbance value of the reaction solution was measured at a wavelength of 620 nm. The soluble sugar content was calculated according to a glucose standard curve.

#### 2.2.4. Assay of Apparent Carotenoids

The total carotenoid content was monitored using the method of Ma et al. [19], with a slight modification. After liquid nitrogen grinding, the mango powder sample (1.0 g) was thoroughly mixed and shaken with 10 mL of acetone in the dark for two hours. After centrifugation at 6000× *g* for 10 min, the supernatant was diluted and measured for the absorbance value at a wavelength of 440 nm. Carotenoid content (mg/100 g) = OD440 nm × 20 (conversion coefficient) × dilution ratio × total volumes of sample extract × 100/pulp quality.

#### 2.2.5. Vitamin C Content Analysis

The vitamin C content was assayed using the method of Thi Thanh Huong et al. [20], with slight amendments. After liquid nitrogen milling, the mango powder sample (6.0 g) was blended with 30 mL of a pre-cooled ethanedioic acid-EDTA solution for 20 min and then centrifuged (10,000× *g*, 20 min, 4 °C). The supernatant was filtered and prepared as the test solution. Next, 8 mL of filtrate, 2 mL of oxalic acid-EDTA solution, 1 mL of three percent metaphosphoric acid-acetic acid, 2 mL of five percent sulfuric acid solution, and 4 mL of five percent ammonium molybdate solution were added to three test tubes, followed by adjustment to 50 mL with distilled water, thorough shaking, and then heating in a 30 °C water bath for 20 min. The reaction solution was measured for the absorbance value at a wavelength of 705 nm. The vitamin C content was calculated referring to an ascorbic acid standard curve.

#### 2.2.6. Measurement of CAT and POD Activity

Hydrogen peroxide clear enzyme activities were measured referring to the method of Bhardwaj et al. [21], with slight modifications. The extraction of the enzyme solution: After liquid nitrogen milling, the mango powder sample (2.0 g) and 10 mL of a pre-cooled phosphoric acid buffer (50 mM, pH 7.0) containing 4% PVP were put into a centrifugal tube, and they were subsequently blended in an ice bath for 20 min. After centrifugation at 3000× *g* for 30 min at 4 °C, the supernatant was collected as a crude enzyme. For the CAT activity, a reaction system, containing 2.8 mL 0.02 M hydrogen peroxide solution, was performed by adding 0.3 mL crude enzyme solution, and it was monitored for the varied absorbance value at a wavelength of 240 nm for 2 min; a change of 0.001 per minute was finally defined as one enzyme activity unit (U). For the POD activity, the reaction system (including 2.7 mL phosphoric acid buffer, 0.3 mL four percents guaiacol solution, and 0.1 mL 0.15 M hydrogen peroxide solution) was processed by adding 0.1 mL crude enzyme solution, and monitored for the varied absorbance value at the wavelength of 470 nm for 3 min, and a change of 0.01 per minute was finally defined as one enzyme activity unit (U).

#### 2.2.7. Assay for SOD Activity

The activity of the SOD enzyme was detected according to the method of Bhardwaj et al. [21], with slight amendments. The extraction of the enzyme solution: After liquid nitrogen milling, the mango powder sample (1.0 g) and 5 mL PBS buffer were added into a centrifugal tube and shaken and extracted for 20 min. After centrifugation at 10,000× *g* for 15 min at 4 °C, the supernatant was collected as a crude enzyme. Seven beakers were prepared for the reaction, namely, three for the treatment, three for the light control, and one for the dark control. Next, 1.5 mL 50 mM PBS buffer, 0.3 mL 130 mM Met solution, 0.3 mL 750 μM NBT solution, 0.3 mL 100 μM EDTA-Na_2_ solution, and 0.3 mL 20 μM riboflavin solution were added to each beaker. Then, 0.5 mL distilled water and 0.1 mL crude enzyme were added to the three treatment groups, while 0.6 mL distilled water was added to the four control groups. In the dark control group, riboflavin was added and immediately placed in the dark to block light. For the other two groups, the breakers were illuminated under a light intensity of 2000 lux for 20 min. All samples were measured for absorbance at a wavelength of 560 nm, and the enzyme required for 50% inhibition of the NBT photochemical reduction by the reaction system per minute was defined as one unit (U) of SOD activity.

#### 2.2.8. Determination of PG Activity

The activity of the PG enzyme was assayed with the method of Deng et al. [22], with some modifications. The extraction of the enzyme solution: After liquid nitrogen grinding, the mango powder sample (6.0 g) and 6 mL 95% ethanol were thoroughly mixed at 4 °C for 10 min. After centrifugation at 8000× *g* for 10 min at 4 °C, the residue was collected and washed with 12 mL of pre-cooled eighty percent ethanol solution, followed by centrifugation again. Then, 10 mL 0.05 M acid acetate–sodium acetate extraction buffer (pH 5.5) was added and gently oscillated until the residue was entirely suspended with ice bath cooling. After centrifugation at 10,000× *g* for 10 min at 4 °C, the supernatant was collected as a crude enzyme solution and evenly divided into active and inactive enzyme solutions, and the latter was boiled for 5 min as a control enzyme sample. Reaction system: 1 mL extraction buffer and 0.5 mL of one percent polygalacturonic acid solution were added to the control and treatment tubes, respectively. Then, 0.5 mL control and active enzyme solutions were added to the treatment and control tubes, respectively, and kept at 37 °C for 1 h. Subsequently, 1.5 mL DNS chromogenic agent was added to both tubes and boiled for 5 min. Finally, the reaction system was diluted to 20 mL and measured at a wavelength of 540 nm for the absorbance value. PG activity was calculated as the mass of galacturonic acid generated from the polygalacturonic acid catalyzed by the enzyme at 37 °C per gram per hour, according to a glucose standard curve.

#### 2.2.9. Measurement of PL Activity

The extraction of the crude enzyme was performed referring to Ge et al.’s methods [23], with some amendments. After liquid nitrogen grinding, the mango powder sample (6.0 g) was mixed with 20 mL 0.05 M Tris-HCL buffer (pH 8.0, containing 1 M sodium chloride). After centrifugation at 10,000× *g* for 20 min at 4 °C, the supernatant was evenly divided into active and inactive enzyme solutions, and the latter was boiled for 20 min as a control enzyme solution. PL activity was measured according to the method of Martin et al. [24], with mild modifications. To the control and treatment tubes, 0.5 mL active enzyme solution and inactive enzyme solution were added, respectively, followed by the addition of 2 mL preheated 0.5% pectin solution, and they were kept at 40 °C for 5 min. Subsequently, 0.5 mL of the reaction solution from the treatment or control tube was rapidly sucked out, and the reaction was stopped by adding 4.5 mL 0.01 M hydrochloric acid solution. PL activity was defined as one enzyme activity unit by the amount of enzyme that the absorbance increased by 1.0 at a wavelength of 235 nm when the sample decomposed pectin, per minute.

#### 2.2.10. Gene expression Analysis

The extraction of RNA was performed referring to the hot boric acid method [25]. Then, protein and DNA pollution was removed with an RNA purification kit (Tiangen Biochemical Technology Co., Ltd., Beijing, China). Finally, RNA was dissolved in sterilized ultrapure water and stored at −80 °C. A cDNA template was synthesized using a TranScript^®^ One-Step gDNA Removal and cDNA Synthesis SuperMix kit (TransGen Biotech Co., Ltd., Beijing, China), and the content was determined using a Nanodrop spectrophotometer (Thermo Scientific™ NanoDrop 2000, Waltham, MA, USA). The related genes and primers were screened via a literature search [26,27,28,29,30,31,32,33,34,35,36], and the primers were synthesized by the company (Invitrogen Trading Co., Ltd., Shanghai, China). All genes and quantitative real-time PCR (qPCR) primers selected in this experiment are shown in Appendix A. A gene expression analysis was performed according to the method of Lawson et al. [37], with slight amendments. To the reaction system, 10 μL TB Green^®^ Premix Ex TaqTM (Tli RNaseH Plus) (2×), 0.4 μL 10 μmol/L forward and reverse primers, 0.4 μL ROX Reference dye II (50×), and 20 ng cDNA template, were added in order, and the volume was finally made up to 20 μL with sterilized ultrapure water. The amplification procedure proceeded as follows: pre-denaturation at 95 °C for 5 min and a total of 40 cycles in succession, including 95 °C for 5 s and 60 °C for 30 s. All samples were prepared in triplicate.

### 2.3. Statistical Analysis of Data

All experiments described were performed in triplicate. The data statistics were completed with Microsoft Excel 2010, and graphs were drawn using Sigmaplot 10.0. SPSS 25.0 (paired-sample *t*-test) was used for a significance analysis. Correlation analysis was conducted with Origin 2021. *p* < 0.05 is labeled as *, and *p* < 0.01 is labeled as **.

## 3. Results and Discussion

### 3.1. The 0.01% Morin Treatment Could Delay the Quality Deterioration of Postharvest Mango Fruits Stored at Ambient Temperature

Flavonoids have been reported for their antioxidant, anti-inflammatory, anti-apoptosis, and free radical scavenging abilities. Interestingly, morin, a common polyhydroxy flavone compound in plants, presents antioxidant and anti-inflammatory pharmacological effects in mammals [38]. Dhanasekar et al. [39] showed that morin had a practical anti-inflammatory function in monosodium urate crystal-induced inflammation in rats. Yang et al. [40] further proved that morin, separated and purified from the methanol extract of mulberry fruit, can be an effective antioxidant and antibacterial substance. Accordingly, morin is natural, safe, and non-toxic. A recent study also showed that morin had a significant preservation effect on postharvest banana fruit, which could inhibit the degradation of cell wall polysaccharides in fruit and delay the ripening and senescence of fruit by regulating the metabolism of nutrients (soluble sugar, amino acids, saponins, etc.) [41]. Moreover, treatment with a mixture of morin and luteolin could not only delay the ripening process of banana fruit but could also reduce the occurrence of collar rot. Since both mango and banana undergo similar postharvest physiological processes, the isologous mechanism of morin in retarding the quality deterioration of postharvest mango fruits should be further elucidated.

Fruit appearance quality is closely related to its commodity value. In this study, the mango peel gradually changed from green to yellow during storage, accumulating browning and disease spots with fruit senescence (Figure 1).

Color difference is one of the critical indicators in evaluating the freshness of fruits. Figure 2A–C show that the luminance (L*) value and the hue angle (H*) value of the fruit pulp decreased with storage. The L* and H* values of 0.01% in the control fruit were lower than those in the morin-treated fruit, with a higher decline rate. However, the chroma (C*) value of the fruit pulp increased in general in the control group, and the variation in the C* value of the 0.01% morin treatment group was lower. Additionally, fruit hardness is also an essential quality index for mango, and it displayed a downward trend as shown in Figure 2D, especially from day 6 to day 12. However, the firmness of the fruits treated with 0.01% morin decreased more slowly than that of the control fruits.

Furthermore, the soluble sugar content contributes to forming fruit flavor, and it increased continuously during storage in the control mango pulp, while the augmentation of the 0.01% morin treatment mango pulp was much lower at the late storage stage (Figure 3A). Therefore, 0.01% morin treatment is favorable for restraining the increase in the soluble sugar content of postharvest mangoes. Carotenoids are natural plant pigments, generally found in yellow, orange, and red plant tissues, and their accumulation is concomitant with fruit maturity. The carotenoid content in the mango pulp increased during storage, which is positively associated with the color change of the mango pulp. At the same time, the 0.01% morin treatment inhibited its accumulation rate in the mango pulp (Figure 3B). Hence, carotenoid synthesis and the flesh-yellowing of mango fruit can be delayed via a 0.01% morin treatment. Vitamin C is also one of the primary nutrients. The vitamin C content in the control group decreased rapidly from day 9 to day 12 during storage. In contrast, the vitamin C content in the mango flesh after the 0.01% morin treatment was generally higher and decreased relatively slowly (Figure 3C). Therefore, the 0.01% morin treatment could inhibit vitamin C degradation in the mango pulp.

Consequently, morin treatment could also delay the ripening and senescence of postharvest mango fruits, as with bananas. The 0.01% morin treatment could delay the color change to yellow and the decrease in the firmness of the mango fruits, retard the ripening and senescence process, restrain the augmentation of carotenoids and soluble sugar, prevent the consumption of vitamin C, and finally sustain the appearance quality.

### 3.2. The 0.01% Morin Treatment Could Enhance the Storage Tolerance of Mango Fruits by Inducing Antioxidant Enzyme Activity

Antioxidant enzymes, including SOD, POD, and CAT in fruits and vegetables, are involved in the metabolic balance of reactive oxygen species (ROS) [42]. SOD can convert active oxygen, such as superoxide anions (O_2_^.-^), into hydrogen peroxide (H_2_O_2_), which then is degraded into water molecules by CAT and POD, thereby reducing the cellular membrane damage caused by free radicals and delaying senescence. Consequently, SOD is a critical antioxidant metalloenzyme in balancing the cellular reactive oxygen metabolism. In the 0.01% morin treatment, SOD enzyme activity in the pulp increased significantly in the middle period of storage (Figure 4A) and was much higher than that in the control, which might indicate a greater O_2_^.-^ production in the control fruits. CAT is also a necessary antioxidant enzyme that scavenges produced H_2_O_2_ to prevent accumulation during metabolism, the activity of which is related to plant stress resistance. However, during the whole period of storage, CAT enzyme activity in the pulp showed an overall increasing trend in the control group, and it had a higher competence than in the 0.01% morin treatment group (Figure 4B), which might indicate more cumulative H_2_O_2_ from the SOD reaction in the initial period of storage. In comparison, the activity of CAT in the 0.01% morin treatment increased slowly, which was probably due to the enhanced SOD activity in the middle period of storage. Additionally, POD is an essential oxidoreductase involved in the stress resistance process and generally has a high activity in senescent tissues, and its activity gradually increased with the storage period but slightly decreased later on. In contrast, the enzyme activity in the 0.01% morin treatment group was significantly lower ((Figure 4C), demonstrating a similar variation trend to CAT. Accordingly, SOD was recognized as a predominant function in the 0.01% morin treatment fruits in the middle period of storage, compared with CAT and POD in the later period. Previous studies found that calcium treatments could not only delay ripening but could also enhance the disease resistance of ‘Keitt’ mango fruits [43]; the calcium treatments similarly promoted antioxidant enzyme activity, including SOD, CAT, and ascorbate peroxidase (APX) activity, in the late storage period of the mangoes and maintained a low ROS content in the pulp tissue. However, another study showed that exogenous melatonin could inhibit ROS accumulation via the enhancement of non-enzymatic (ascorbic acid and glutathione content) and enzymatic (glutathione reductase activity) antioxidants and, thus, maintain the quality and storage life of ‘Guiqi’ mango fruits [44]. Therefore, the antioxidant effect could differ across mango varieties. It was speculated that the 0.01% morin treatment could eliminate excessive O_2_^.-^ by enhancing SOD enzyme activity in the middle period of storage and then inducing an increase in CAT and POD enzyme activity during anaphase storage, preventing the accumulation of H_2_O_2_ in vivo and, thus, maintaining the balance of the cellular ROS metabolism and ultimately helping in delaying the deterioration in quality of mango fruits. The probable signaling regulation of H_2_O_2_ produced by SOD in the 0.01% morin treatment mango fruits should be further discussed.

### 3.3. The 0.01% Morin Treatment Could Retard the Softening of Mango Fruits by Suppressing Cell Wall-Degrading Enzyme Activities

The plant cell wall is mainly composed of pectin and cellulose. PG and PL act as degraders of pectin, thus initiating fruit softening. In this study, the PG enzyme activity of the mango fruits displayed a decreasing trend during storage (Figure 5A). Specifically, a significant enhancement from day 6 to day 9 occurred in the control fruits. In contrast, the enzyme activity in the 0.01% morin treatment group maintained a decreasing and lower trend throughout the storage period. It was hypothesized that the softening of the mango fruits might be related to the reinforcement of PG activity and that the 0.01% morin treatment delayed the disintegration of the cell wall structure by inhibiting an increase in PG activity and, finally, delaying fruit senescence. PL is also a typical pectinase that catalyzes the elimination cleavage of pectin molecular chains, and its overall trend displayed an increase in the control fruits during storage, as shown in Figure 5B. Nevertheless, the enzyme activity in the 0.01% morin treatment was markedly lower, with a catastaltic decline from day 6 to day 9. It was presumed that the morin treatment could suppress the activity of the PL enzyme to maintain fruit firmness. A previous study showed that heat treatment at 50 °C could mitigate the activities of PG, pectin methylesterase (PME), and PL enzymes in banana fruits; slow down the degradation of pectin substances; and extend shelf life [45]. Ali et al. also found that carboxymethyl cellulose (CMC) treatment could delay the softening of mango fruits, and their cellulase, PME, and PG enzyme activity were significantly reduced [1]. Therefore, the 0.01% morin treatment could reduce the PG and PL enzyme activity of mango fruits, inhibiting pectin degradation and delaying the softening process.

### 3.4. The 0.01% Morin Treatment Might Prolong the Storage Period of Mango Fruits by Delaying the Expression of Ripening-Related Genes

A decline in firmness, color transformation, soluble solid accumulation, sweetness augmentation, and aroma component synthesis in mango fruits during storage are closely related to the expression of critical genes in metabolic pathways, such as ethylene signaling, cell wall-degrading enzymes, carotenoid synthesis, and starch conversion.

Ethylene, as a phytohormone, is a prerequisite for initiating the ripening and senescence of fruits and vegetables. 1-Aminocyclopropane-1-carboxylic acid synthase (ACS) and 1-aminocyclopropane-1-carboxylic acid oxidase (ACO) are two critical enzymes in ethylene biosynthesis, and their activities determine ethylene production [46]. Ethylene can bind to ethylene receptors on the plasma membrane, which can induce the expression of cell wall-degrading enzymes, carotenoid synthesis, starch conversion, and even disease resistance-related genes through ethylene signaling pathways. Therefore, ethylene signal transduction can be directly regulated by ethylene synthesis and receptor gene expression to delay or accelerate the ripening and senescence of fruit. In the current study, *ACS* and *ACO* expressions increased sharply in the late storage stage and were positively correlated with the ripening of the mango fruits, which was suppressed in the 0.01% morin treatment group (Figure 6A,B). Meanwhile, ethylene receptor genes (*ETR1*) and ethylene-responsive sensor (*ERS1*) expressions were consistent with the *ACS* and *ACO* genes (Figure 6C,D). In addition, the expression of the ethylene-insensitive gene (*EIN2*), as a positive regulator in the ethylene signaling pathway, increased rapidly on day 9 of storage (Figure 6E). The ethylene response factor (*ERF1*) gene expression was also significantly increased (Figure 6F). However, in the treatment group, *ETR1*, *ERS1*, *EIN2*, and *ERF1* were uniformly suppressed. Zaharah et al. [47] showed that nitric oxide (NO) fumigation treatment could delay ‘Kensington Pride’ mango fruits’ ripening and softening, reducing ethylene synthesis by inhibiting the activity of ACS and ACO. Furthermore, Winterhagen et al. [48] revealed that the mango ethylene receptor genes *ETR1* and *ERS1* played a regulatory role in fruit abscission and maturation. Hong et al.’s [32] further studies showed that pre- and post-harvest salicylic acid (SA) and NO treatments inhibited *ACO* and *ERS1* expressions but increased *ETR1* and *EIN2* expressions, thereby prolonging the shelf life of mangoes. Accordingly, it was speculated that the 0.01% morin treatment could inhibit *ACS* and *ACO* expressions to reduce the synthesis of ethylene; retard the expressions of *ETR1*, *ERS1*, *EIN2*, *ERF1*, and other genes in the ethylene signal transduction pathway, thus negatively regulating the gene expression of different ethylene response metabolic pathways; and, finally, delay the ripening and senescence of the fruits.

A decline in firmness is a crucial quality deterioration of postharvest fruits regulated by ethylene. Studies have shown that the transcription and activity of nine PG enzyme family members in mangoes are positively correlated with fruit ripening [49]. At the same time, the PG gene promoter sequence contains binding elements of transcription factors, including MADS-box, EIN3, and ERFs, indicating that the ethylene signal transduction pathway regulates the transcription of PG genes. In addition, a decline in the firmness of mango flesh can be accelerated via the increasing gene expressions of lipoxygenase (*LOX*), expansin protein (*EXP*), endonuclease 1,4-β-glucanase (*EG*), and β-galactosidase (*βGal*). LOX can destroy the cell membrane structure through a membrane lipid peroxidation reaction and participate in ethylene synthesis. EXP can loosen the cell wall structure and contribute to cell wall degradation. The glycosidase activity of βGal can modify cell wall pectin substances and promote the disintegration of the cell wall structure. EG is a hemicellulase that also participates in cell wall disintegration. Therefore, the *LOX*, *EXP*, *βGal*, and *EG* expressions in mangoes enhance with storage time and are negatively correlated with fruit firmness. In this study, all expressions increased significantly at day 6 in the control group, while they increased at day 9 in the treatment group (Figure 7). Chidley et al. [31] found that ethylene treatment significantly increased *βGal* gene expression and accelerated mango flesh softening, and Zhang et al. [50] showed that 6-Benzylaminopurine (6-BA) treatment inhibited ‘Guifei’ mango LOX enzyme activity and downregulated *MiLOX* gene expression, thus retarding the ripening process. 1-MCP treatment can delay the softening of mangoes by inhibiting the expression of the *MiCel1* gene and increasing EG enzyme activity [51]. Zheng et al.’s [29] research showed that oxalic acid treatment not only inhibited αGal (α-galactosidase), βGal, and PG enzyme activity but also inhibited *EXP* gene expression and delayed the fruit softening process. In this study, the 0.01% morin treatment delayed the peak expression levels of the *LOX*, *EXP*, *EG*, and *βGal* genes in mango fruits by 3 days (Figure 7). Therefore, it was speculated that the 0.01% morin treatment could generally inhibit ethylene-regulated pathway transcriptions, delaying the increase in *LOX*, *EXP*, *βGal*, and *EG* gene expressions; retarding cell wall degradation; and, thus, slowing down fruit softening.

Carotenoid synthesis is directly responsible for the color change of mango flesh. Phytoene synthase (PSY) and Lycopene β-cyclase (LCYB), two key enzymes in the carotenoid synthesis pathway, determine the accumulation of carotenoids in mango flesh during ripening. During the storage of mango fruits under ambient conditions, *PSY* expression showed an overall enhancing trend, and *LCYB* expression revealed an increasing and then decreasing trend (Figure 8A,B). Interestingly, the reinforcement of *PSY* expression in the 0.01% morin treatment group lagged behind that in the control group. Moreover, the peak of the *LCYB* gene expression in the control group occurred on day 6, while that in the 0.01% morin treatment group occurred on day 9. Xia et al. [52] showed that the accumulation of carotenoids increased with *PSY* gene expression according to fruit color. Karanjalker et al. [53] found that a high *LCYB* expression could cause an anthocyanin accumulation both in yellow and red mangoes during maturation. However, Ma et al. [19] indicated that *PSY* expression was upregulated with the maturation of mangoes, while *LCYB* expression was inhibited in ‘Tainong1′ and ‘Hongyu’ mango cultivars. Therefore, the gene expression regulation of pigment synthesis could differ across mango cultivars. Nevertheless, the change in the carotenoid content and firmness were positively correlated, as well as the ethylene production in mango fruits [54]. Therefore, it was speculated that the 0.01% morin treatment could inhibit the expressions of the *PSY* and *LCYB* genes through the ethylene signal transduction pathway, retard the synthesis of carotenoids and other pigments, and delay the yellowing of the pulp.

During ripening, the starch in mango fruit converts into soluble sugar, including sucrose, fructose, and glucose, and SPS is a critical enzyme in the process of sucrose synthesis. In our study, *SPS* expression firstly increased and then decreased rapidly with the extension of storage time, while that in the 0.01% morin treatment was significantly lower and suppressed to some extent (Figure 8C), which is consistent with the tendency of soluble sugar content. In previous studies, it was shown that a decreased SPS enzyme activity in ‘Irwin’ mangoes was associated with sucrose accumulation during mango growth [55], and a decrease in SPS enzyme activity could delay the softening and ripening process in ‘Jinhwang’ mangoes [56]. Therefore, SPS enzyme activity directly affects the maturation of mangoes. Additionally, Sun et al. [57] found that, in apples during storage, Harvista (a commodity name for the 1-MCP reagent, which can competitively bind to the ethylene receptor) treatment could effectively inhibit the degradation of starch; delay the rise in soluble sugar, reducing sugar, sucrose, glucose, and fructose content; and restrain the increase in *SPS* enzyme activity and gene expression. Therefore, it was speculated that the 0.01% morin treatment could prevent the accumulation of soluble sugar and decelerate starch conversion in the mango flesh by retarding the ethylene signal transduction to inhibit the expression of the *SPS* gene, finally resulting in quality maintenance during storage.

In addition, UCP can obstruct the synthesis of adenosine triphosphate (ATP) by uncoupling the electronic respiratory chain, which intensifies energy dissipation, leads to a decrease in the intracellular energy charge (EC) value, and accelerates the aging process in fruits and vegetables. In the present study, the *UCP* expression in mango fruits was enhanced with the extension of storage time and peaked on the 9th day, while that in the 0.01% morin treatment maintained a lower expression level (Figure 8D). A previous study found that the ATP content and EC value decreased significantly during the postharvest senescence of litchi fruits, revealing a negative relationship with the browning index [58]. A former study revealed that the expressions of the alternate oxidase (*AOX*) and *UCP* genes were closely related to mango ripening. Among them, *UCP* was enriched in the color transfer period, and *AOX* was increased in the mature period [34]. A further study found that the enhanced expression of tomato *AOX* and *UCP* genes during fruit ripening might be associated with increased respiration in the late period and could be influenced by ethylene signaling molecules [59]. Therefore, it was assumed that the 0.01% morin treatment may inhibit the synthesis of ethylene, thereby reducing the expression of the *UCP* gene and reducing the energy loss in the stored mangoes, thus slowing fruit ripening and senescence.

Mango fruits may be infected by pathogens after harvest, and the transcription of disease resistance-related genes, such as phenylalanine ammonia lyase (PAL), functionating in the biosynthesis of phenylpropanoids, can help to improve storage tolerance. Among them, phenylpropanoid metabolites can produce flavonoids through the flavonoid pathway. Additionally, chalcone synthase (CHS) plays an important role in plant disease resistance through the flavonoid synthesis pathway; when a pathogen infects a plant, chitosan treatment can promote *PAL* and *CHS* expressions, which trigger the biosynthesis of antifungal metabolites, so that the fruit can resist the attack of plant pathogenic fungi [60]. In mango fruits, *PAL* and *CHS* gene expressions increased sharply in the late storage period but remained at a low level in the treatment group (Figure 9A,B). Additionally, polyphenol oxidase (PPO) and peroxidase (POD) can oxidize polyphenols into quinones, participate in the process of tissue corkification and lignification, and play an essential role in the pathogen infection of plant tissues. Under ambient conditions, the *PPO* and *POD* gene expressions in the mangoes were enhanced with storage time. Nonetheless, the gene expressions of *PPO* and *POD* were lower in the treatment group, and the rise in the *PPO* gene expression lagged behind that of the control group (Figure 9C,D). In addition, β-1,3-glucanase (GLU) and chitinase (CHI) are important disease-related proteins involved in plant defense responses to pathogens. *GLU* and *CHI* gene expressions increased notably in the middle and late storage periods of the mangoes, indicating that the fruits began to suffer from pathogenic bacteria, while the increases in *GLU* and *CHI* gene expressions in the treatment group lagged behind those in the control group (Figure 9E,F). Previous research found that heat treatment combined with UV-C irradiation enhanced the expressions of *PAL*, *POD*, *GLU*, *CHI*, and other genes in stored mangoes and effectively suppressed anthrax [61]. Another study found that benzothiadiazole (BTH) treatment notably elevated the expression levels and enzyme activity of the *PPO* and *POD* genes in ‘Keitt’ and ‘Zill’ mangoes, increased the total phenol content, and induced disease resistance during fruit storage [35]. However, the transcriptions of *PAL*, *CHS*, *PPO*, *POD*, *GLU*, and *CHI* were not upregulated after the 0.01% morin treatment (Figure 9). The above results indicate that no direct regulation effect on the primary disease resistance-related genes in mango could be induced by the 0.01% morin treatment, which could provide a defense against pathogenic microorganisms by delaying fruit ripening and senescence.

Finally, a correlation analysis was further applied to reveal the interaction between nutritional quality, antioxidant and cellwall-degrading enzyme activity, and ripening-related genes (Figure 10). During the ripening, the luminance, hug angle, and firmness in mango fruits showed a negative correlativity with ethylene synthesis and signal transduction pathways genes’ expression. Moreover, the activity of PG and SOD also presented a generally negative correlation with those gene expressions, which indicates their functions in the early stage of storage for mango fruits. However, there was a significant positive correlation between ethylene-regulated pathways gene expression and ripening-related gene expressions, including cell wall degradation, carotenoid synthesis, starch conversion, energy metabolism, and even disease resistance. Therefore, it was speculated that morin may delay the ripening and senescence of mango fruits through the ethylene-regulated pathways.

## 4. Conclusions

In summary, morin treatment can delay the ripening and senescence of postharvest mango fruits. Our results showed that a 0.01% morin treatment could mitigate the decrease in mango firmness during storage at ambient temperature (25 ± 1 °C), slow down the decrease in L* and H* values, and increase the C* value to preserve fruit appearance quality. Meanwhile, a 0.01% morin treatment could delay soluble sugar and carotenoid synthesis and reduce vitamin C consumption, thereby delaying a deterioration in fruit quality. After 12 days of storage, the content of soluble sugar and carotenoids in the treatment group were 68.54 mg/g and 11.20 mg/100 g, respectively, lower than those in control, while the vitamin C content in the treatment group was 0.58 mg/g, higher than that in control. Moreover, a 0.01% morin treatment may remove excessive ROS by enhancing SOD enzyme activity (an over 1.5-fold increase compared with control) and then inducing reinforced CAT and POD enzyme activity to degrade H_2_O_2_ in vivo, ensuring the metabolic balance of ROS in cells. Furthermore, suppressed PG and PL enzyme activity (an over 1-fold decrease compared with control) could help to prevent the rapid degradation of the cell wall structure, thus improving the storability of fruits. An advanced gene expression analysis illustrated that a 0.01% morin treatment could inhibit the expressions of the *ACS* and *ACO* genes and the *ETR1*, *ERS1*, *EIN2*, and *ERF1* genes in the ethylene signal transduction pathway; delay the transcriptions of cell wall-degrading enzymes (*LOX*, *EXP*, *βGal*, and *EG*), carotenoid synthesis key enzymes (*PSY* and *LCYB*), *SPS*, *UCP*, and other ethylene-regulated genes; and eventually retard fruit ripening and senescence. Nevertheless, the expressions of resistance-related genes (*PAL*, *CHS*, *PPO*, *POD*, *GLU*, and *CHI*) were not induced by a 0.01% morin treatment. However, some issues were not resolved in this study. Further research should focus on the transcriptional regulation of morin in the ethylene signal transduction pathway and the relationship between H_2_O_2_ content and mango fruit resistance, because of the notable effect of SOD without CAT and POD. Accordingly, a 0.01% morin treatment can delay the ripening and senescence of zill mangoes and maintain their storage quality, which presents specific application prospects for mango fruit preservation.

## Figures and Tables

**Figure 1 foods-12-04251-f001:**
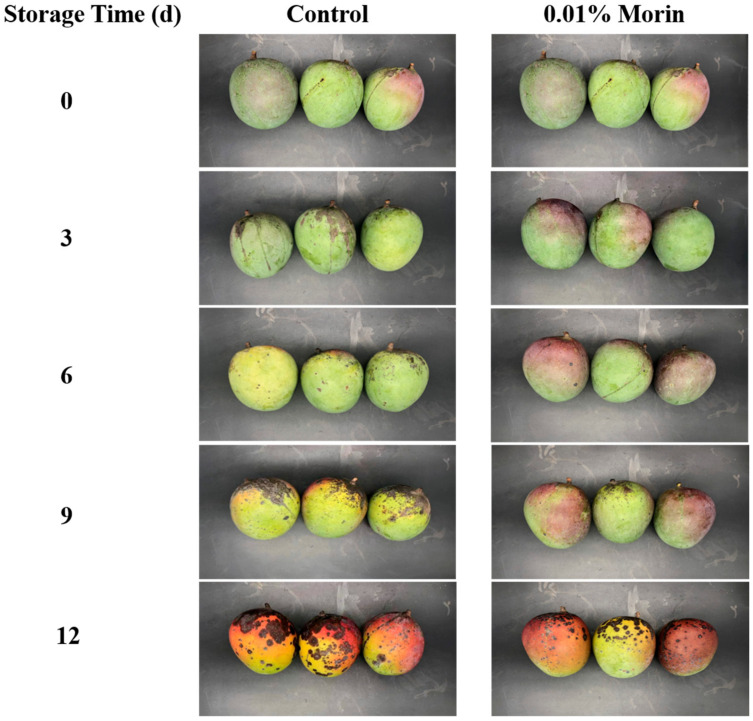
The appearance changes of mango fruits treated with 0.01% morin stored at 25 ± 1 °C for 0, 3, 6, 9, and 12 days.

**Figure 2 foods-12-04251-f002:**
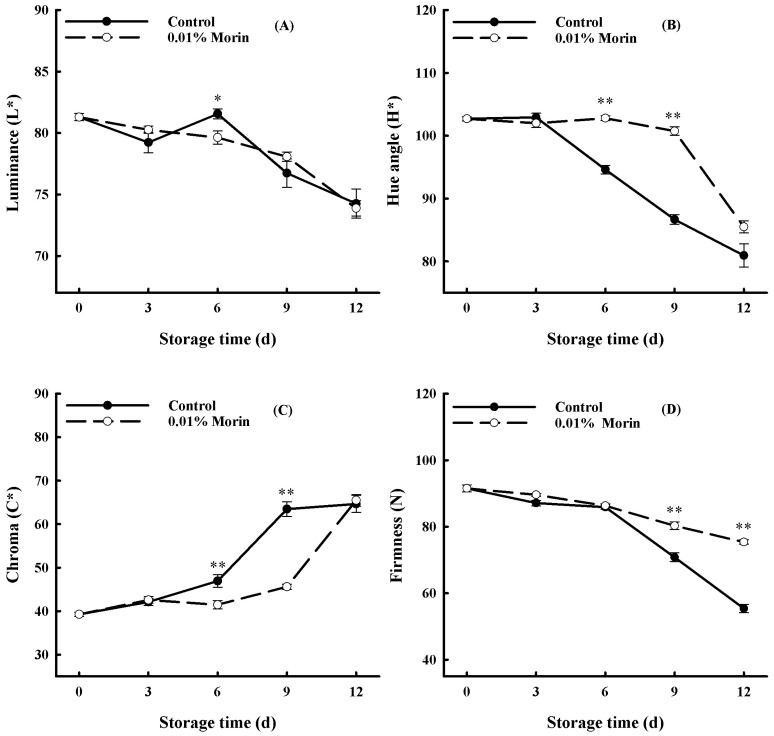
Effects of 0.01% morin treatment on color difference and firmness of mango fruit during storage at 25 ± 1 °C. (**A**) Luminance (L*). (**B**) Hue angle (H*). (**C**) Chroma (C*). (**D**) Firmness. Each value represents the means ± SE of three replicates. Asterisks indicate significant differences (* *p* < 0.05, ** *p* < 0.01) between the 0.01% morin treatment group and the control at each time point of storage.

**Figure 3 foods-12-04251-f003:**
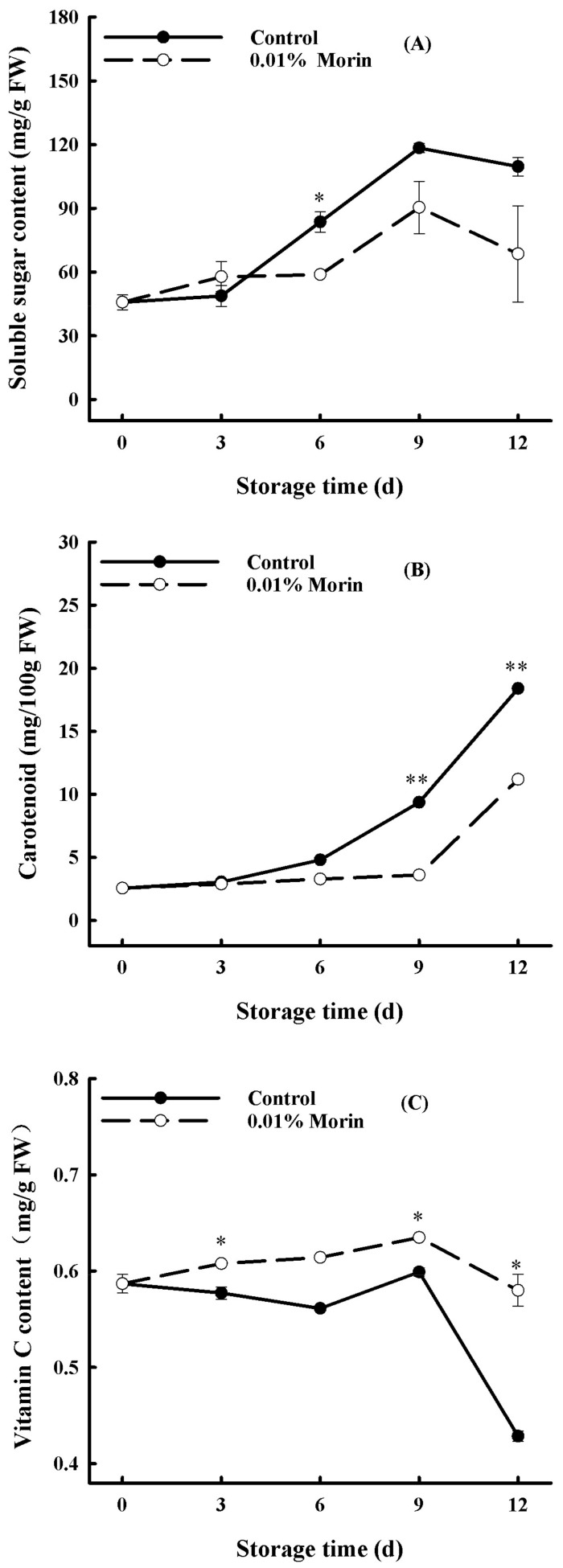
Effect of 0.01% morin treatment on nutritional quality of mango fruits during storage at 25 ± 1 °C. (**A**) Soluble sugar content. (**B**) Carotenoid content. (**C**) Vitamin C content. Each value represents the means ± SE of three replicates. Asterisks indicate significant differences (* *p* < 0.05, ** *p* < 0.01) between the 0.01% morin treatment group and the control at each time point of storage.

**Figure 4 foods-12-04251-f004:**
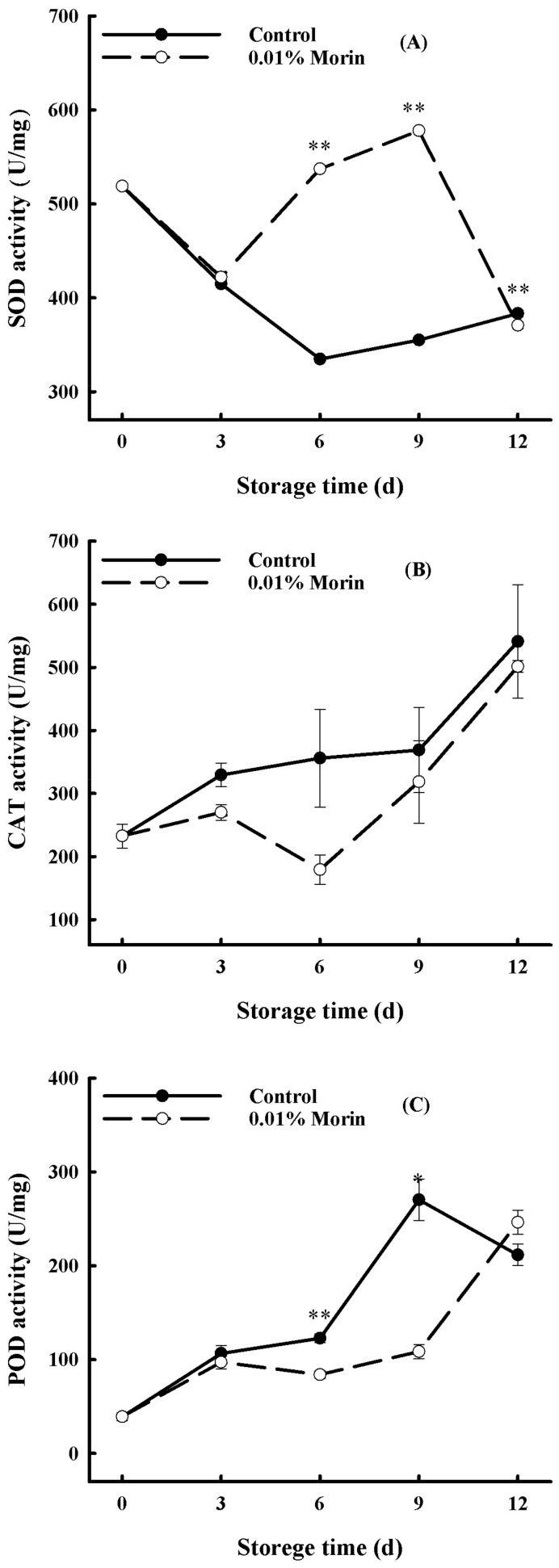
Effect of 0.01% morin treatment on antioxidant enzyme activity of mango fruit during storage at 25 ± 1 °C. (**A**) Superoxide dismutase (SOD) activity. (**B**) Catalase (CAT) activity. (**C**) Peroxidase (POD) activity. Each value represents the means ± SE of three replicates. Asterisks indicate significant differences (* *p* < 0.05, ** *p* < 0.01) between the 0.01% morin treatment group and the control at each time point of storage.

**Figure 5 foods-12-04251-f005:**
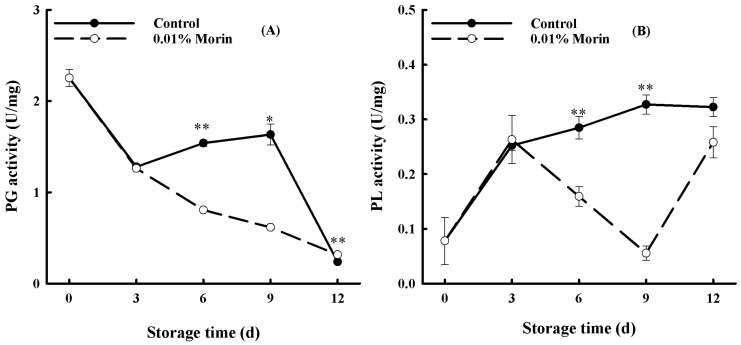
Effects of 0.01% morin treatment on cell wall-degrading enzyme activity of mango fruits during storage at 25 ±1 °C. (**A**) Polygalacturonase (PG) activity. (**B**) Pectate lyase (PL) activity. Each value represents the means ± SE of three replicates. Asterisks indicate significant differences (* *p* < 0.05, ** *p* < 0.01) between the 0.01% morin treatment group and the control at each time point of storage.

**Figure 6 foods-12-04251-f006:**
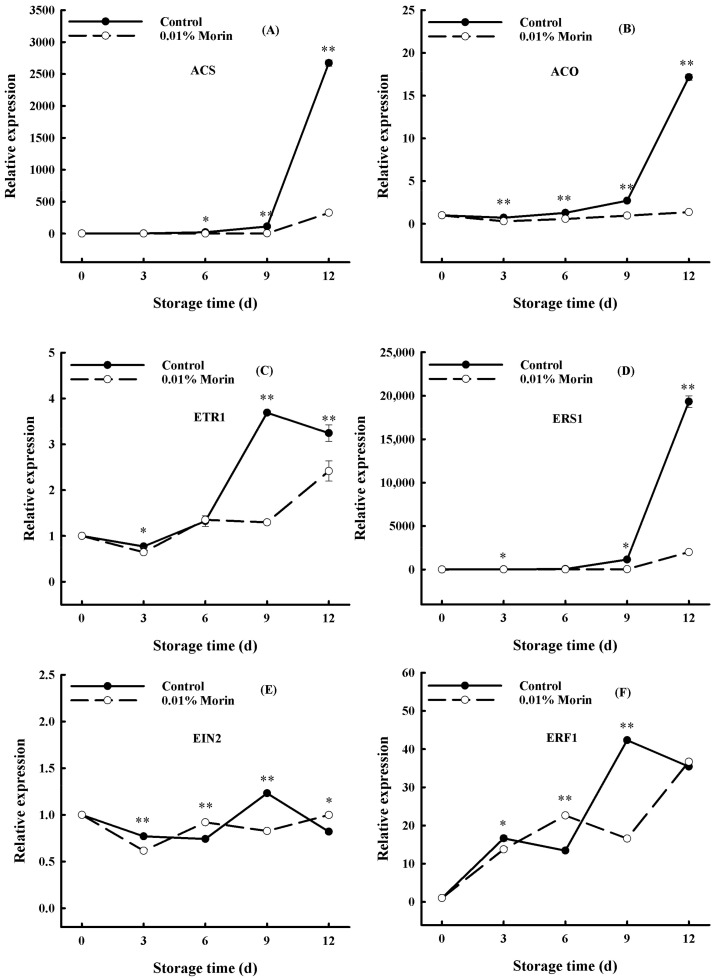
Effects of 0.01% morin treatment on the expression of ethylene biosynthesis-related genes in mango fruit during storage at 25 ± 1 °C. (**A**) 1-aminocyclopropane-1-carboxylic acid synthase (*ACS*). (**B**) 1-aminocyclopropane-1-carboxylic carboxylic acid oxidase (*ACO*). (**C**) Ethylene receptor gene (*ETR1*). (**D**) Ethylene response sensor (*ERS1*). (**E**) Ethylene-insensitive gene (*EIN2*). (**F**) Ethylene response factor (*ERF1*). Each value represents the means ± SE of three replicates. Asterisks indicate significant differences (* *p* < 0.05, ** *p* < 0.01) between the 0.01% morin treatment group and the control at each time point of storage.

**Figure 7 foods-12-04251-f007:**
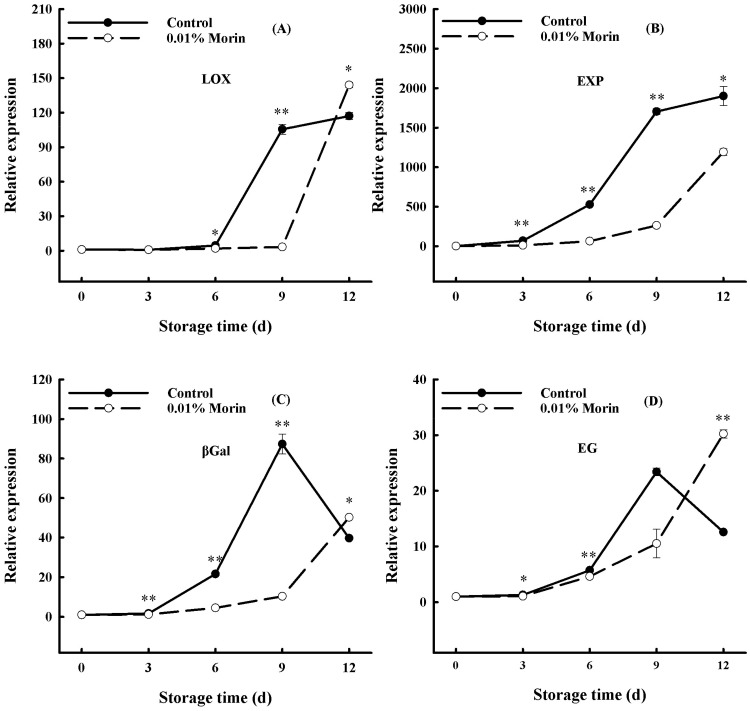
Effects of 0.01% morin treatment on the expressions of genes related to cell wall degradation pathways during storage of mango fruits at 25 ± 1 °C. (**A**) Lipoxygenase (*LOX*). (**B**) Expansion protein (*EXP*). (**C**) β-galactosidase (*βGal*). (**D**) Endoglucanase (*EG*). Each value represents the means ± SE of three replicates. Asterisks indicate significant differences (* *p* < 0.05, ** *p* < 0.01) between the 0.01% morin treatment group and the control at each time point of storage.

**Figure 8 foods-12-04251-f008:**
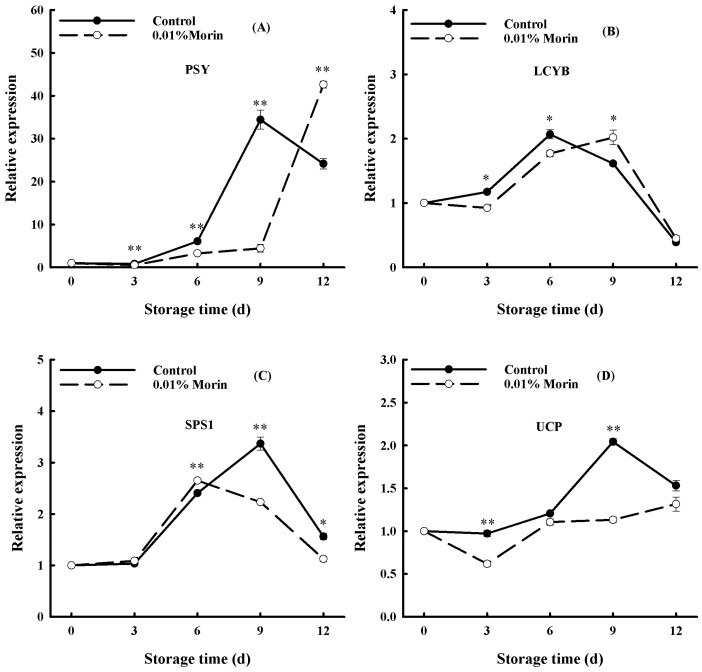
Effects of 0.01% morin treatment on carotenoid synthesis, starch conversion, and energy metabolism pathway-related enzyme gene expressions of mango fruit during storage at 25 ± 1 °C. (**A**) Phytoene synthase (*PSY*). (**B**) Lycopene β-cyclase (*LCYB*). (**C**) Sucrose phosphate synthase (*SPS1*). (**D**) Uncoupling protein (*UCP*). Each value represents the means ± SE of three replicates. Asterisks indicate significant differences (* *p* < 0.05, ** *p* < 0.01) between the 0.01% morin treatment group and the control at each time point of storage.

**Figure 9 foods-12-04251-f009:**
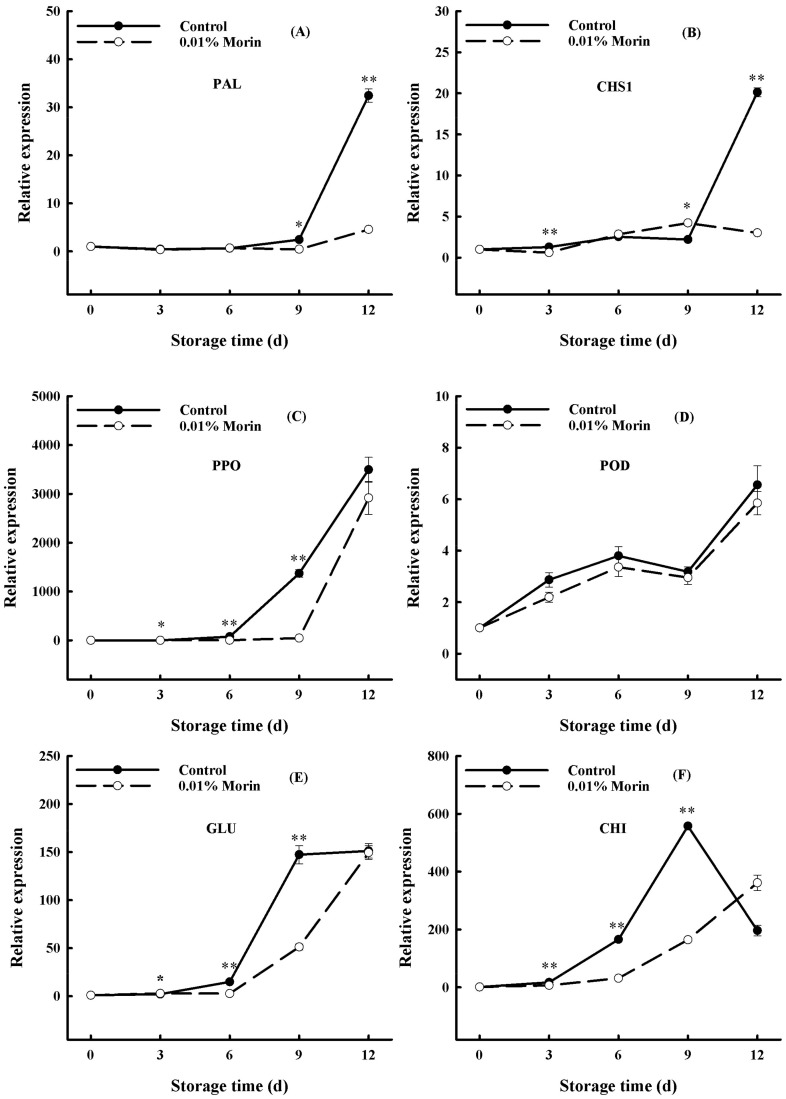
Effects of 0.01% morin treatment on the expressions of disease resistance-related genes in mango fruits during storage at 25 ±1 °C. (**A**) Phenylalanine Ammonia-Lyase (*PAL*). (**B**) Chalcone synthase (*CHS1*). (**C**) Polyphenol oxidase (*PPO*). (**D**) Peroxidase (*POD*). (**E**) β-1,3-Glucanase gene (*GLU*). (**F**) Chitin synthase (*CHI*). Each value represents the means ± SE of three replicates. Asterisks indicate significant differences (* *p* < 0.05, ** *p* < 0.01) between the 0.01% morin treatment group and the control at each time point of storage.

**Figure 10 foods-12-04251-f010:**
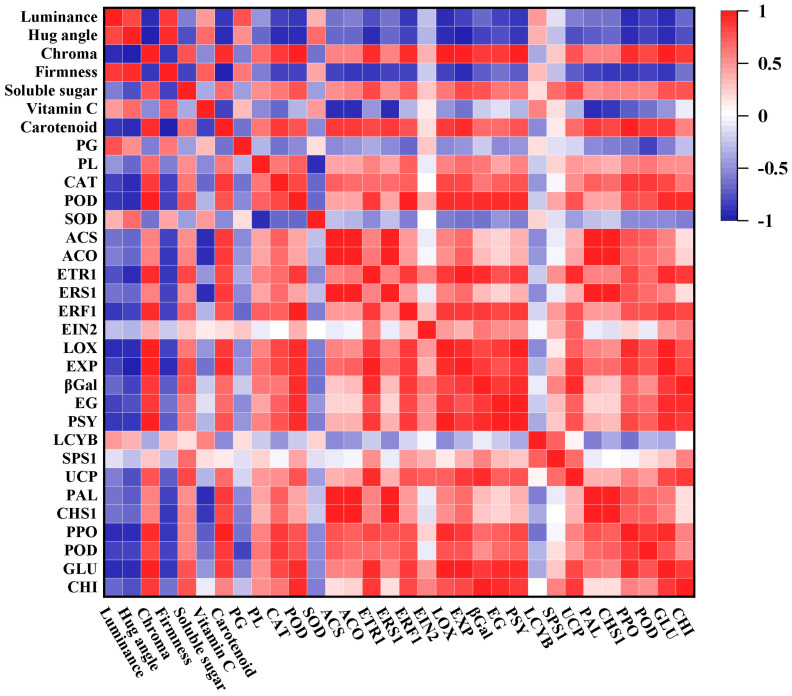
Correlation analysis of 0.01% morin treatment on mango quality, enzyme activity, and ripening-related gene expressions.

## Data Availability

Data is contained within the article or Appendix A.

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
