# Peer review of "Morin Treatment Delays the Ripening and Senescence of Postharvest Mango Fruits"

_foods, 2023, doi:10.3390/foods12234251_

Round 1

Reviewer 1 Report

Comments and Suggestions for Authors

The manuscript is interesting. However, it needs revision before being published. 

Paragraph 2.1. Sampling should be detailed: total number of samples, temperature and conditions of shipping

Paragraph 2.2.4 At the wavelength for the determination of total carotenoids there can be interferences of many other non-carotenoid acetone soluble  compounds. Carotenoids should have been determined by a simple HPLC procedure that is considerably more accurate. The paragraph should be renamed as "Assay of apparent carotenoids".

Paragraph 2.3. A multivariate method, such as principal component analysis or cluster analysis should be applied to the results.

Paragraph 3.1. All the text from "Postharvest fruits....should be further elucidated" is an paragraph that should be moved to "Introduction" (paragraph 1), because the given information has nothing to do with the results obtained by the authors. Authors should avoid non-scientific claims, such as anticancer activity that have not been demonstrated in vivo, so far. All the activities described in the first paragraph of paragraph 3.1. are potential and authors should avoid mention them, if possible. 

Paragraph 3.1. "Furthermore, the soluble sugar content.....nutritional quality". Nutritional quality refers to many nutritional compounds, not only vitamin C. Authors should replace "nutritional quality" by "vitamin C". In this paragraph, authors should better explain the effects of the morin.

Finally, it is widely known that many phenolic compounds have strong flavours. Authors should carry out a simple but sound sensory analysis explaining the possible influence of morin on the mango fruits flavours.

Comments on the Quality of English Language

No comment

Author Response

Dear Reviewer,

We sincerely appreciate your valuable comments on our article, which are greatly helpful in improving our manuscript. We have made an effort to improve the manuscript and made some red-marked changes in the revised manuscript, but it will not affect the content and framework of the paper. The responses of point by point can be found following this cover letter. We express our sincere gratitude for your enthusiastic work and hope our revisions will be recognized. Thank you again for your feedback and suggestions.

The Responses to The Reviewer:

  1. Paragraph 2.1. Sampling should be detailed: total number of samples, temperature and conditions of shipping.

We have carefully reviewed our experimental records and added detailed information on the total sample size, temperature, and transportation conditions in section 2.1. "Fruit Materials and Methods" of the article. Please check that in our revised manuscript.

  1. Paragraph 2.2.4 At the wavelength for the determination of total carotenoids there can be interferences of many other non-carotenoid acetone soluble compounds. Carotenoids should have been determined by a simple HPLC procedure that is considerably more accurate. The paragraph should be renamed as "Assay of apparent carotenoids".

We agree with this much clearer description strongly and have replaced "Assay of total carotenoids" with "Assay of apparent carotenoids" based on your suggestion.

  1. Paragraph 2.3. A multivariate method, such as principal component analysis or cluster analysis should be applied to the results.

Thank you for your valuable advice. We have realized the importance of multivariate methods applied to analyze results. Although our experiment did not directly apply principal component analysis or cluster analysis, we have added a correlation analysis in Paragraph 3.4. in our revised manuscript.

  1. Paragraph 3.1. All the text from "Postharvest fruits.....should be further elucidated" is a paragraph that should be moved to "Introduction" (paragraph 1), because the given information has nothing to do with the results obtained by the authors. Authors should avoid non-scientific claims, such as anticancer activity that have not been demonstrated in vivo, so far. All the activities described in the first paragraph of paragraph 3.1. are potential and authors should avoid mention them, if possible.

Thanks for your great suggestion, we have moved the section of "postharvest fruits..... quality deterioration" in paragraph 3.1 to the introduction section and added reference [3] for clarification. At the same time, we appreciate your detailed pointing out and apologize for our carelessness. Based on your feedback, we have corrected the unscientific expression in lines 246-248 of paragraph 3.1. Please check these in our revised manuscript.

  1. Paragraph 3.1. "Furthermore, the soluble sugar content.....nutritional quality". Nutritional quality refers to many nutritional compounds, not only vitamin C. Authors should replace "nutritional quality" by "vitamin C". In this paragraph, authors should better explain the effects of the morin.

We believe this is a good suggestion, and the description related to nutritional quality in paragraph 3.1. " Furthermore, the soluble sugar content.....nutritional quality" has been deleted. The effects of the morin on postharvest mango fruit were emphasized in the last paragraph in the 3.1 section.

  1. Finally, it is widely known that many phenolic compounds have strong flavours. Authors should carry out a simple but sound sensory analysis explaining the possible influence of morin on the mango fruits flavours.

Thank you for your valuable feedback. In the previous experiments, we also conducted preliminary sensory evaluations on the mango fruits both in the blank group and the experimental group and found that morin treatment did not have a significant impact on the flavor or taste of the fruits after our degustation. Regrettably, the experimental design and processing methods in our study were mainly referred to according to the references [15] and [41], so we have not conducted further sensory analysis. Thanks for your suggestion again, and we will carry out a sound sensory analysis in future work.

Reviewer 2 Report

Comments and Suggestions for Authors

1.      The abstract is too informative. Add more numerical data. List all other treatments in the abstract part.

2.      Why authors have taken mango as a research material (add latest citation).

3.      Lien 128: The vitamin C content was assayed using the method of Thi Thanh Huong et al. [18] With slight amendments. What amendments did the authors make?

4.      Same for Line 141: with slight modifications?

5.      PG and PL activity mean?

6.      In Figure 2 add error bar.

7. Other researchers have already carried out the same type of work. What is the novelty of this work?

8.      In the whole manuscript, authors have given citations from the years 19945 and 2002. Authors are advised to add the latest references.

9.      Line 19-20 Rewrite the line with the latest references.

10.  Rewrite the conclusion part and include numerical data.

Comments on the Quality of English Language

1.      The abstract is too informative. Add more numerical data. List all other treatments in the abstract part.

2.      Why authors have taken mango as a research materials (add latest citation).

3.      Lien 128: The vitamin C content was assayed using the method of Thi Thanh Huong et al. [18] with slight amendments. What amendments authors did?

4.      Same for Line 141: with slight modifications?

5.      PG and PL activity Means?

6.      In figure 2 add error bar.

7.      Same type of works already carried out by other researchers. What is the novelty in this work?

8.      In whole manuscript authors have given citation from the year 19945 and 2002. Authors are advised to add latest references.

9.      Line 19-20 Rewrite the line with latest references.

10.  Rewrite the conclusion part and include numerical data.

Author Response

Dear Reviewer,

We sincerely appreciate your valuable comments on our article, which are greatly helpful in improving our manuscript. We have made an effort to improve the manuscript and made some red-marked changes in the revised manuscript, but it will not affect the content and framework of the paper. The responses of point by point can be found following this cover letter. We express our sincere gratitude for your enthusiastic work and hope our revisions will be recognized. Thank you again for your feedback and suggestions.

The Responses to The Reviewer:

  1. The abstract is too informative. Add more numerical data. List all other treatments in the abstract part.

We sincerely appreciate your valuable feedback. To meet the requirement of a word number in the abstract (200 words maximum), we have rewritten the abstract and added some numerical data to better present our research methods and results, based on your feedback. Moreover, correlation analysis was also applied. Please check that in our revised manuscript. Thanks for your patient review again.

  1. Why authors have taken mango as a research material (add latest citation).

We have carefully reviewed our article and added two recent references in the introduction section of our article, namely references [2] and [3], which could provide us with more in-depth research background and theoretical support to better explain why we used mangoes as research materials. Please check that in our revised manuscript.

  1. Line 128: The vitamin C content was assayed using the method of Thi Thanh Huong et al. [18] With slight amendments. What amendments did the authors make?

Thank you very much for your careful attention and patient review. In the analysis of vitamin C content in paragraph 2.2.5., we made the following modifications: Firstly, we made adjustments to some of the reagents and the operating sequence of the experiment based on the method of Thi Thanh Huong et al [20]. Specifically, our experimental design did not select bromine water, thiourea, and 2,4-Dinitrophenylhydrazine (DNPH solution) reagents, and correspondingly added oxalic acid- EDTA and ammonium molybdate solution. Secondly, we adjusted the measurement wavelength and chose a wavelength of 705 nm to measure our final sample.

  1. Same for Line 141: with slight modifications?

Thank you for your question. In paragraph 2.2.6., we referred to the method of Bhardwaj et al. [21] and made the following modifications to the determination of catalase (CAT) and peroxidase (POD) activity: Firstly, we selected phosphate buffer PBS containing 4% PVP to extract the enzyme solution, which was extracted on ice for 20 minutes. The mixture was centrifuged at 4℃, 3,000×g, and for 30 minutes. Secondly, in the process of measuring POD activity, we selected a wavelength of 470 nm to measure our final sample. Finally, the activity units of CAT and POD were expressed as U/mg.

  1. PG and PL activity mean?

Thank you for your question. PG and PL mean polygalacturonase and pectin lyase, respectively, which are the two main enzymes involved in the softening of fruit, through degrading the cell wall.

  1. In Figure 2 add error bar.

Thank you very much for pointing out the issue of adding error bars to Figure 2 in our article. We have carefully considered your suggestion and apologize for our carelessness. This is partly due to our small error values, which makes some of the error lines in the figure unclear enough. We have now adjusted the error lines in Figure 2 and made them bold.

  1. Other researchers have already carried out the same type of work. What is the novelty of this work?

Although other studies have already done similar work, our research still contains some innovations as follows. Firstly, data on the mechanism of morin delaying fruit ripening and senescence is still limited, especially since its effects on mango fruit have not yet been reported. Secondly, although there are two references to the preservation effect of banana fruit after harvest, the mechanism of delaying the ripening and senescence at the gene expression level by morin treatment still needs to be explored. Finally, our study further elucidates the effect of morin on delaying the ripening process of mango fruit from the perspective of molecular biology, which emphasizes the role of the ethylene regulatory pathway in morin treatment compared with previous reports.

  1.  In the whole manuscript, authors have given citations from the years 19945 and 2002. Authors are advised to add the latest references.

Thank you for pointing out our requirement to add the latest references. We have carefully reviewed our references again. There are three citations between the years 1994 and 2002. Firstly, in the gene expression analysis of section 2.2.10., our method for extracting RNA referred to the hot boric acid method proposed by Wan et al. [25] in 1994. At present, this literature has been cited multiple times by a large number of researchers and is a great classic reference. Secondly, in our experiment, the design of quantitative real-time PCR (qPCR) primers for uncoupling protein (UCP) genes was based on the study by Considine et al. [34] in 2001. Finally, the design of peroxidase (POD) and polyphenol oxidase (PPO) gene expression primers was based on the study of Lin et al. [35] in 2001. Therefore, the cited literature above has been fully validated and widely applied and is reliable for application in our study.

  1. Line 19-20 Rewrite the line with the latest references.

We apologize for our confusion with this suggestion. Because line 19-20 belongs to the section of the abstract in our manuscript, and generally speaking, references should not be allowed in the abstract. Nevertheless, we have cited some latest references for the analysis of the activities of antioxidant and softening-related enzymes in the result.

  1. Rewrite the conclusion part and include numerical data.

Thank you for your review and valuable feedback on the conclusion part of our manuscript. Based on your suggestion, we have rewritten the conclusion section and added some relevant experimental numerical data to enhance the credibility and persuasiveness of the results.

Round 2

Reviewer 2 Report

Comments and Suggestions for Authors

The Author properly incorporates all the suggestions. In this version, the author has given more importance to technical accuracy and hence the manuscript may be accepted.